

# The influence of storage time on ponazuril concentrations in feline plasma

Sherry Cox[1], Lainey Harvill[1], Sarah Singleton[1], Joan B. Bergman[1] and Becky DeBolt[2]

[1] Department of Biomedical and Diagnostic Sciences, College of Veterinary Medicine, University of Tennessee, Knoxville, TN, United States of America
[2] Department of Small Animal Clinical Sciences, College of Veterinary Medicine, University of Tennessee, Knoxville, TN, United States of America

## ABSTRACT

**Background**. The pharmacokinetics of ponazuril have been determined in several species; however, there is very little information on the stability of the drug after storage for long periods of time. This study was undertaken to determine the stability of ponazuril in plasma samples stored at −80 °C, which is the temperature most commonly used in the author's laboratory.

**Method**. Spiked plasma samples (0.3, 7.5, and 15 µg/mL) were stored at −80 °C for three months. Analysis occurred on the first day and then once a week for the following twelve weeks. The drug was extracted using a chloroform extraction and separated by high performance liquid chromatography using ultraviolet detection.

**Results**. There was no loss of drug for any concentration for the first four weeks of storage. There was an average loss of less than 5% from day 35 through day 70 and an average loss of 6% on day 77 and 84. The data suggest that ponazuril is stable for 4 weeks when stored at −80 °C and undergoes minimal loss in the remaining 8 weeks.

## INTRODUCTION

Ponazuril (also known as toltrazuril sulfone) is from the triazine class of antiprotozoal drugs and is a metabolite of the drug toltrazuril (*Love et al., 2015*). Triazine compounds have been used in veterinary medicine to treat the harmful effects of protozoa such as *Toxoplasma gondii*, *Coccidia* spp., and *Isopora*, (*Stock, Elazab & Hsu, 2018*).

The use of drugs in different species requires knowledge of their pharmacokinetics. An understanding of the pharmacokinetics is necessary for determining the most appropriate dose, route of administration and dosing interval. The determination of accurate drug concentrations is extremely important in the estimation of pharmacokinetic parameters and may impact design of the drug regimen. To determine these properties, it is essential to calculate the concentration of the compound of interest after administration. Because of the importance placed on the validation of bioanalytical methods that are used in the determination of data for pharmacokinetic studies the FDA has established guidelines (FDA, Guidance for industry: Bioanalytical Method Validation). The information in

Corresponding author
Sherry Cox, scox6@utk.edu

the guidelines applies to bioanalytical procedures such as chromatographic methods. A necessity for the generation of the data is a validated analytical procedure. However, if proper conditions are not maintained for storage the results may not accurately indicate the sample concentration at the time it was determined. For most studies, samples are stored until there is an adequate quantity for analysis, which requires them to be stored, usually frozen for a period of time. To guarantee accurate results it is essential to determine drug degradation from samples using the storage conditions anticipated for the study.

Pharmacokinetic studies of ponazuril have been determined in several species (*Laczay, Voros & Semjen, 1995*; *Epe et al., 2005*; *Ghanem et al., 2008*; *Love et al., 2015*; *Prado et al., 2011*; *Dirikolu et al., 2011*; *Zou et al., 2014*; *Stock, Elazab & Hsu, 2018*; *Furr & Kennedy, 2020*. To the author's knowledge there are no other stability articles involving drugs used in felines or triazine drugs. Thus, the intention of this research was to determine if concentrations of ponazuril in feline plasma decreased after storage for 12 weeks at −80 °C using a validated method (*Cox et al., 2021*).

## MATERIALS & METHODS

### Equipment

The equipment used for ponazuril assessment consisted of a 2695 separation module and a 2487 ultraviolet detector (Waters Milford, MA, USA). Separation was achieved on a Waters Symmetry Shield $RP_{18}$ (4.6 mm × 150 mm, 5 μm) column. The mobile phase consisted of 0.1% formic acid in water and acetonitrile (50:50, v/v). The ultraviolet detector was set at 254 nm and the flow rate was 1.1 mL/min.

### Reagent and solutions

Ponazuril and diclazuril were purchased from Cayman Chemical (Ann Arbor, MI, USA). Both compounds had a purity greater than 99%. All other solvents and chemicals were acquired from Fisher Scientific (Pittsburg, PA, USA). Stock solutions of ponazuril (100 μg/mL) and diclazuril (100 μg/mL), which is the internal standard, were prepared by dissolving appropriate amounts of each compound in methanol. Cat plasma was fortified with ponazuril to construct daily standard curves (0.1–25 μg/mL). These were accepted if the coefficient of determination ($r^2$) was greater than 0.99.

### Sample collection

Six adult cats from UTCVM faculty and staff were determined to be healthy based on results of physical examination and history were used. Healthy animals are routinely used in pharmacokinetic studies in order to prevent distortion of results due to abnormal metabolism, disease processes or interference from drugs the animal is taking. Venous blood was collected from the jugular vein of each cat and placed in lithium heparin tubes. The volume of blood collected from each cat did not exceed 1% of body weight in kilograms. Enough collected blood was centrifuged to provide plasma for the study. Plasma was pooled into one container and then divided into three separate tubes, and then spiked with one of the ponazuril calibration standards (0.3, 7.5, and 15 μg/mL) that were used in the study. Aliquots from each of the three individual standards were then placed in separate

vials labeled Day 1, 7, 14, 21, 28, 35, 42, 49, 56, 63, 70, 77, and 84 for analysis on those dates. These concentrations were chosen as low medium and high standards within the linear curve used. Samples were then placed in the −80 °C. Day 1 samples were analyzed on that day. Because of animal size and availability enough plasma was obtained to analyze only one sample for each ponazuril concentration evaluated.

### Extraction method

Ponazuril was extracted from plasma samples that were thawed one time using a liquid-liquid extraction method (*Cox et al., 2021*). Frozen plasma samples were thawed at room temperature (22 °C) for ∼30 min, vortex-mixed, and 100 μl of plasma was transferred to a screw top tube followed by 10 μl of diclazuril (100 μg/mL, internal standard) and two mL chloroform. The tubes were placed on a rocker for 15 min and then underwent centrifugation for 20 min at 1,000× g. The organic layer was transferred to a glass tube and evaporated to dryness with nitrogen gas. Samples were reconstituted in 250 μL of mobile phase and 100 μL was analyzed.

### Statistical analysis

Comparative changes in the actual values of different measured ponazuril concentrations in plasma samples were used to summarize the effect of storage length on stability of the drug.

## RESULTS

Samples were analyzed on days 1, 7, 14, 21, 28, 35, 42, 49, 56, 63, 70, 77, and 84 (Figs. 1A–1D and Table 1). The actual values of the different measured ponazuril concentrations and the percent variation of the concentration compared to the starting concentration are reported in Table 1. Ponazuril concentrations were determined in all samples. There was no loss of drug for any concentration (low, medium or high) for the first four weeks (days 1 through 28) of storage at −80 °C. Please refer to the Figure and Table for the remainder of the results.

The ponazuril method of analysis that was used for the stability study was previously validated (*Cox et al., 2021*). The method produced a linear curve over the concentration range used (0.1–25 μg/ml) with a lower limit of quantification of 0.1 μg/mL. The intraday and interday variability ranged from 3.7 to 10% and 2.1 to 5.5% while the average ponazuril recovery was 99% and the average recovery of diclazuril was 94%. Calibration curves were analyzed each day of plasma analysis.

## DISCUSSION

In order to determine the optimal storage time for ponazuril samples at −80 °C, plasma was collected from healthy felines and pooled in order to reduce the effects of inter-individual variability. A validated method of analysis was applied to the samples to determine the effect of storage duration. Plasma samples spiked with various amounts of ponazuril that fell within a validated linear concentration range and stored in a −80 °C freezer for 84 days

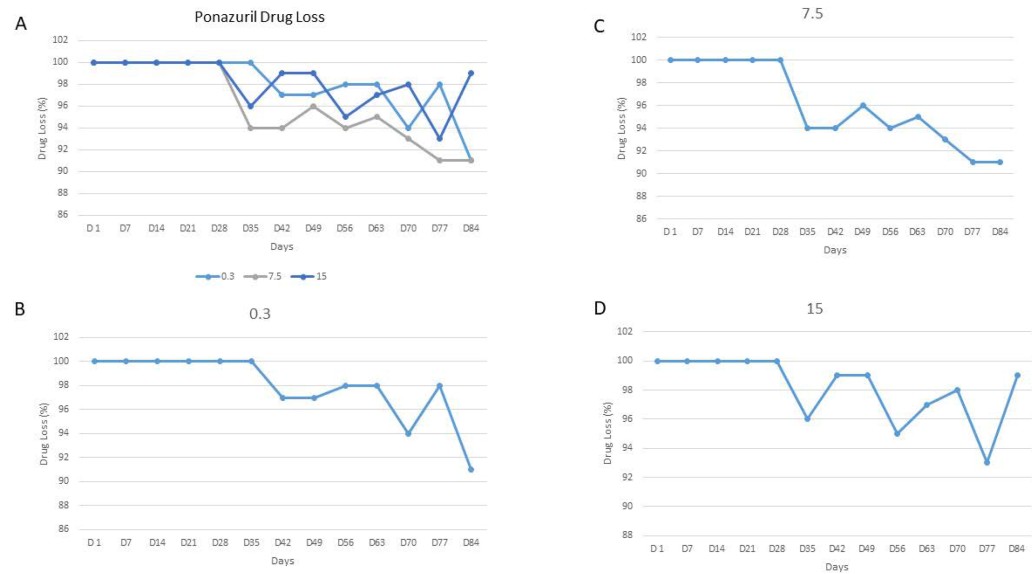

**Figure 1** **Ponazuril Drug Loss.** Percent ponazuril loss in plasma over 84 days: (A) all three concentrations, (B) 0.3 µg/mL, (C) 7.5 µg/ml, (D) 15 µg/ml.

**Table 1** **Ponazuril stability information for 84 days after storage at −80 °C.**

| Starting conc. | Concentration at different days of storage | | | | | | | | | | | | |
|---|---|---|---|---|---|---|---|---|---|---|---|---|
| | Day 1[a] | Day 7 | Day 14 | Day 21 | Day 28 | Day 35 | Day 42 | Day 49 | Day 56 | Day 63 | Day 70 | Day 77 | Day 84 |
| 0.3 | 0.3 | 0.3 | 0.3 | 0.3 | 0.3 | 0.3 | 0.291 | 0.291 | 0.294 | 0.294 | 0.282 | 0.294 | 0.273 |
| 7.5 | 7.5 | 7.5 | 7.5 | 7.5 | 7.5 | 7.05 | 7.05 | 7.2 | 7.05 | 7.1 | 7.0 | 6.8 | 6.8 |
| 15 | 15.0 | 15.0 | 15.0 | 15.0 | 15.0 | 14.4 | 14.85 | 14.85 | 14.25 | 14.55 | 14.7 | 13.95 | 14.85 |
| **Percentage change in concentration from starting concentration** | | | | | | | | | | | | | |
| 0.3 | 0 | 0 | 0 | 0 | 0 | 0 | −3 | −3 | −2 | −.2 | −6 | −2 | −9 |
| 7.5 | 0 | 0 | 0 | 0 | 0 | −6 | −6 | −4 | −6 | −5 | −6 | −9 | −9 |
| 15 | 0 | 0 | 0 | 0 | 0 | −4 | −1 | −1 | −5 | −3 | −2 | −7 | −1 |

**Notes.**

Concentration results reported in µg/mL; $n = 1$.

[a]Day 1 samples were not frozen but analyzed immediately after spiking.

were used. One limitation to the study was a lack of duplicate samples for the analysis. The analysis of only one sample did not allow for any statistical analysis.

Ponazuril was detected in all concentrations for the entire 84 days. For the first 28 days, there was no change in any of the concentrations. Indicating that the drug is stable after storage for that length of time. After 28 days, there was a slight loss of drug for all three concentrations. The average loss was 4% or less for days 35 through 63, which would still not have an impact on higher concentrations and have a minimal impact on lower concentrations. The samples should still be viable for this period of time. On day 70, there was an average loss of 5% and on days 77 and 84 there was an average loss of 6%. The loss

on day 84 was greater for the 0.3 and 7.5 than the 15 µg/mL standard. Overall, the average loss from day 7 though day 84 was 2%, 4% and 2% for 0.3, 7.5 and 15 µg/mL, respectively.

   To the author's knowledge there are no other stability articles involving drugs used in felines or triazine drugs. There is a paucity of information in the literature about the storage stability of drugs in biological samples in general. Determination of the concentrations in biological samples is fundamental to describing pharmacokinetic parameters which in turn leads to appropriate dosing regimens. Therefore, it is imperative to determine the effect that storage can have on drug stability.

## CONCLUSIONS

The intention of this study was to determine the effect of storage duration at −80 °C on the stability of ponazuril concentrations in plasma. In summary, plasma samples, containing ponazuril can be stored for 28 days with no loss of drug. By applying the FDA Bioanalytical Guidelines (*FDA, 2018*) for method development that states a value should be within 15% deviation from the standard all of the samples were stable for 84 days when stored at −80 °C. This study provides much needed information on the stability of ponazuril plasma samples stored at −80 °C.

### Funding
The authors received no funding for this work.

### Competing Interests
The authors declare there are no competing interests.

### Author Contributions
- Sherry Cox conceived and designed the experiments, performed the experiments, analyzed the data, prepared figures and/or tables, and approved the final draft.
- Lainey Harvill and Sarah Singleton performed the experiments, authored or reviewed drafts of the paper, and approved the final draft.
- Joan B. Bergman performed the experiments, analyzed the data, authored or reviewed drafts of the paper, and approved the final draft.
- Becky DeBolt conceived and designed the experiments, authored or reviewed drafts of the paper, and approved the final draft.

### Data Availability
   The data is available in the Supplementary Files.

### Supplemental Information
Supplemental information for this article can be found online at http://dx.doi.org/10.7717/peerj.12516#supplemental-information.

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
