# Peer review of "The influence of storage time on ponazuril concentrations in feline plasma"

_PeerJ, doi:10.7717/peerj.12516_

## Round 0.1 · original submission · Major Revisions

Please respond to each of the reviewers' comments with an explanation for how you modified the manuscript to resolve them. Please do not only refer to changes on a particular line(s) of the manuscript - copy and paste the actual changes made in your response letter to the reviewers.

·

Basic reporting

This is a straight-forward and readable article that contributes to the literature on measuring drug concentrations over time in individual animals. The importance of the selected drug, ponazuril, is highlighted, and the rationale for sample storage is included. It might be helpful to also include how important accurate drug concentrations can be to the estimation of accurate pharmacokinetic parameters, which may impact drug regimen design. Other context might also be provided in the introduction or methods by referencing the FDA guidelines there, in addition to the already included reference in the discussion.

The included figure is somewhat helpful, and it might be more helpful to use linear graphs, rather than bar charts, given the expectation that concentrations over time will decay in a linear or log linear manner.

Reference to literature on other drug decay over time after storage might be helpful to provide context to these findings, either by relevant chemical class, or drugs in this species of animals. If there is no relevant literature, noting this might also provide context.

Experimental design

The design is for the most part clearly explained. It might be more clear if the following were addressed:

1. Consider providing a brief explanation of why plasma was collected from healthy animals (i.e., as part of the description of this step in the study).
2. Re-word the description of plasma spiking and aliquoting into vials. It was not initially clear to me that 3 different concentrations would be followed throughout the study.
3. Considering explaining why duplicate or triplicate spiked plasma samples were not created for each day, and why duplicate or triplicate sampling of each vial was not performed - this would have provided a bit more confidence in the estimated concentrations.
4. Include a brief overview of the analytical method herein, even though it is a previously described method, perhaps at least the instrumentation and basic outline.
5. Include at what temperature the vials were thawed - was it room temperature? How long did it typically take for thawing before samples were analyzed?
6. Include perhaps parenthetically that the diclazuril was added as the internal standard.
7. Consider re-wording the description of the numeric results, or perhaps consider a table with the actual concentrations along with the percent losses.

Validity of the findings

The conclusions are valid based on the data, and clear interpretation of what these data mean is provided. As previously mentioned, it would be helpful to put these data and the conclusions in the context of similar drugs in other species, or other drugs in cats, in order to highlight the importance of knowing how much storage impacts drug concentrations and how that impact affects pharmacokinetic parameter estimates (and therefore drug dosing recommendations). In addition, comments on whether or not there is a contribution of this study to the expectation of linearity of drug degradation during storage could be interesting.

Additional comments

I appreciate the authors making the effort to evaluate and publish these kind of storage data - this is an understudied (or at least under-reported) area in clinical pharmacokinetics in veterinary medicine.

Reviewer 2 ·

Basic reporting

see comments below

Experimental design

See comments below

Validity of the findings

See comments below

Additional comments

Comments for Authors:

The Introduction is a little long. There is no need to review past pharmacokinetic studies or what it is used for. Try and shorten the introduction and focus on just why this study was performed.

On line 52 is says “possible drug loss”. Try another term here. Perhaps “degradation of the drug” or “loss of stability”.

Line 58-59: Be more specific here. What animal species? What concentration range? Storage time? What was the duration and conditions of storage to be examined?

Line 69: It says “Ponazuril and diclazuril were purchased from Cayman Chemical”. For a study like this you really should be using a USP analytical reference standard. (obtained from www.USP.org) If you didn’t use that, then there has to be some assurance that this source was equivalent to a USP standard, or met acceptance criteria from the USP. The acceptance criteria for each of these standards is listed in the USP monograph.

Line 73-78: This section describes the HPLC method. For a study such as this, which focuses on the integrity of the method, you need a validated HPLC method. Therefore, you need to consult with reputable sources (ICH, USP, FDA, etc.) and perform the necessary validation steps. At the least, you must establish LOQ and LOD (and specify the criteria), repeatability of the assay, and performance of the calibration curve. How many points in the calibration curve were used? Were all points on the linear calibration curve?

I realize that on line 109 you said that a validation was performed. But you should have stated that earlier (in the Introduction). During the validation study that was cited on this line, the stability tests and effects of storage should have been performed as part of the validation. Thus this study may have been unnecessary. Usually authors are discouraged from splitting a single study into two publications.
Line 78: You must specify the anticoagulant used. The same anticoagulant must be used for all samples throughout the study.

Line 86: Ideally in a study like this, at least 3 replicates per sample should be used. You need this to calculate statistical parameters. This is an important shortcoming of the study.

Line 87: Were freeze-thaw effects measured? What was the extraction efficiency?

Line 104-108: Put all your specific results in a table and just refer to the table. It is difficult to read when presented in the text.

Line 125: Here you describe the loss of drug and amounts. But what was your acceptance criteria? At the beginning of the study (planning stages) you should have had some acceptance criteria listed (eg, loss of 5%, 10%, 15%, etc.). What is acceptable before you reject the samples at these storage times? This must be stated. It is preferable that you use some published and accepted guidelines for this decision (ie, validation guidelines published by ICH, FDA, etc.) This information doesn’t appear until the last line on line 139. This should have been an objective stated at the beginning of the paper.

Figure 1: Title says “percent recovery”. This is not the correct term to use here. Percent strength of solution or something similar is preferred. Us the full axis for presentation. (0% to 110%)

---

## Round 0.2 · Minor Revisions

Please read the comments from Reviewer 2 carefully, and develop a response to them, including any changes that you make in the manuscript to address them.

·

Basic reporting

No comment.

Experimental design

No comment.

Validity of the findings

No comment.

Additional comments

I am satisfied with revisions the authors have made.

Reviewer 2 ·

Basic reporting

I did not fill in each box in this review. That would be rather tedious and redundant. Instead just scroll down to my comments below.

Experimental design

I did not fill in each box in this review. That would be rather tedious and redundant. Instead just scroll down to my comments below.

Validity of the findings

I did not fill in each box in this review. That would be rather tedious and redundant. Instead just scroll down to my comments below.

Additional comments

The authors have submitted a revision and addressed most, but not all of the concerns expressed by the reviewers.
This paper is acceptable as is, but has some significant limitations and short-comings that need to be stated in the final version. They did not perform a full validation, and although we understand the limitations, many readers might assume that this paper constitutes a validation of their method. The lack of replicates, among other criteria results in shortcomings.

The main criticism that still remains is that the authors performed a study without prior acceptance criteria to meet and a hypothesis to fulfill. They should have stated at the beginning what their goals were. They should have acceptance criteria for storage of the samples. Do stored samples meet these criteria?
It is a simple question. Ordinarily +/- 10% would be an acceptable goal. (This is common for many USP standards and guidelines.) You met a goal in this range, so why not state this range as your acceptance?

---

## Round 0.3 · accepted · Accept

I appreciate your taking the reviewers' comments to heart, and using them to improve your manuscript.